# MESHGEN: GENERATING PBR TEXTURED MESH WITH RENDER-ENHANCED AUTO-ENCODER AND GENERATIVE DATA AUGMENTATION

## ABSTRACT

In this paper, we present MeshGen, an advanced image-to-3D pipeline designed to generate high-quality 3D objects with physically based rendering (PBR) textures. Existing methods struggle with issues such as poor auto-encoder performance, limited training datasets, misalignment between input images and 3D shapes, and inconsistent image-based PBR texturing. MeshGen addresses these limitations through several key innovations. First, we introduce a render-enhanced point-to-shape auto-encoder that compresses 3D shapes into a compact latent space, guided by perceptual loss. A 3D-native diffusion model is then established to directly learn the distribution of 3D shapes within this latent space. To mitigate data scarcity and image-shape misalignment, we propose geometric alignment augmentation and generative rendering augmentation, enhancing the diffusion model's controllability and generalization ability. Following shape generation, MeshGen applies a reference attention-based multi-view ControlNet for image-consistent appearance synthesis, complemented by a PBR decomposer to separate PBR channels. Extensive experiments demonstrate that MeshGen significantly enhances both shape and texture generation compared to previous methods.

## 1 INTRODUCTION

With the rapid advancement of diffusion-based image generation models, there has been significant progress in automatic 3D generation. In particular, methods utilizing score distillation sampling (Poole et al., 2023) have demonstrated breakthroughs by leveraging priors from text-to-image diffusion models. However, these optimization-based methods are relatively slow and face challenges such as mode collapse (Wang et al., 2023a;c) and the Janus problem (Armandpour et al., 2023; Seo et al., 2023) due to the lack of inherent 3D information. Subsequent strategies address these challenges by focusing on multi-view generation (Liu et al., 2023c; Long et al., 2023; Chen et al., 2024c; Voleti et al., 2024) and large reconstruction models (Zou et al., 2023; Tang et al., 2024a; Hong et al., 2023; Li et al., 2023a; Xu et al., 2024b; Liu et al., 2024; Wei et al., 2024). The former generates multi-view images for 3D reconstruction. The latter maps sparse view images to compact 3D representations using neural networks, such as triplane NeRF (Chan et al., 2021) or grid 3D Gaussians (Zou et al., 2023; Tang et al., 2024a). While these methods have improved the quality and speed of 3D generation, they typically use volumetric representations such as NeRF or Gaussian instead of 3D meshes, resulting in further loss of quality during conversion (Hong et al., 2023; Chen et al., 2024b; Tang et al., 2023). Moreover, these methods, which rely solely on render loss for supervision, are highly susceptible to inconsistencies across multiple synthesized views and often struggle to reconstruct objects with complex geometric structures (Sun et al., 2024).

Recently, 3D native diffusion methods have garnered significant attention as a promising paradigm towards mesh-oriented generation (Gupta et al., 2023; Wang et al., 2023b; Zhang et al., 2024a; Li et al., 2024b; Wu et al., 2024b; Hong et al., 2024; Chen et al., 2024a). By mapping 3D meshes into a compact latent space using 3D auto-encoders, these methods directly learn the distribution of 3D shapes instead of reconstructing from generated multi-views. Despite considerable progress has been made, several challenges remain unresolved. Firstly, the inherent limitations of current 3D auto-encoders preclude the integration of perceptual loss during training, leading to less detailed reconstructed meshes and consequently constrained expressiveness of the latent space. Moreover,

existing 3D native diffusion methods typically generate simple and symmetric shapes, making it challenging to match the input images, and the scarcity and poor quality of public datasets further limit the generalization ability of current open-source models. In addition, existing image-guided texture generation methods struggle to produce appearances consistent with the original images and can only generate materials with light baked in, rather than the physically based rendering (PBR) materials required in practical applications.

In response to these challenges, we introduce MeshGen, a novel image-to-3D pipeline specially designed to generate PBR textured meshes that closely resemble the provided image in both geometry and appearance. Specifically, to enhance the expressiveness of the point-to-shape auto-encoder, we propose a triplane-based auto-encoder that incorporates perceptual render loss during training, thereby fully exploiting the memory efficiency of triplane compared to latent vector set representation. Next, based on the geometrical covariant property of the point-to-shape auto-encoder and the appearance-invariant nature of image-to-shape diffusion, we establish an image-to-shape diffusion model with geometric alignment and generative rendering augmentation to enhance image-shape consistency and generalization ability. For texture generation, we propose using a geometry-conditioned ControlNet with reference attention fine-tuning to generate multi-view images consistent with the input image in both appearance and lightning. We then employ a PBR decomposer to estimate the PBR components in the shaded image and a texture inpainter to fill in the invisible parts. As a result of these advancements, MeshGen can generate PBR textured 3D assets with consistent geometry and exceptional fidelity within 30 seconds.

To summarize, our contributions are:

- We propose the MeshGen auto-encoder, which substantially improves the expressiveness of the point-to-shape auto-encoder by incorporating both geometric and appearance supervision. It utilizes a coarse-to-fine optimization strategy guided by render-based perceptual loss, ensuring more accurate shape representation.
- We introduce a novel image-to-shape pipeline with our proposed geometric alignment augmentation and generative rendering augmentation, which largely enhance image-shape alignment and generalization capabilities.
- We design a reference attention-based image-conditioned mesh texturing pipeline. Coupled with our proposed PBR decomposer, our method is capable of generating relightable textures that closely align with the appearance of the input image.

## 2 RELATED WORK

### 2.1 3D GENERATION

Early efforts in 3D generation focus on per-scene optimization methods based on CLIP similarity (Radford et al., 2021; Sanghi et al., 2021; Jain et al., 2022) and score distillation sampling (Poole et al., 2023). By utilizing powerful pre-trained image diffusion models, these methods soon excel in various 3D generation tasks (Wang et al., 2023c; Chen et al., 2023b; Lin et al., 2023; Tang et al., 2023; Chen et al., 2024b; Shi et al., 2023b; Li et al., 2023c; Wang & Shi, 2023; Sun et al., 2023; Chen et al., 2023c). Despite great success has been achieved, optimization-based methods still suffer from slow generation speed and low success rates. To overcome these challenges, researchers have explored multi-view generation (Liu et al., 2023b; Tang et al., 2024b; Lu et al., 2023; Liu et al., 2023c; Long et al., 2023; Wu et al., 2024a; Li et al., 2024a; Chen et al., 2024c; Voleti et al., 2024) and large reconstruction models (Szymanowicz et al., 2023; Liu et al., 2023e; Xu et al., 2023; 2024a; Hong et al., 2023; Li et al., 2023b;a; Tang et al., 2024a; Wang et al., 2024). InstantMesh (Xu et al., 2024b) adopts a two-stage optimization strategy that firstly trains a multi-view to triplane NeRF model, then uses this model as initialization for FlexiCubes (Shen et al., 2023), thus yielding direct textured mesh reconstruction from images. MeshLRM (Wei et al., 2024) follows a similar pipeline but uses differentiable marching cubes with deferred rendering for direct mesh output. MeshFormer (Liu et al., 2024) utilizes a hierarchical voxel structure for efficient large reconstruction model training. Although these methods have advanced 3D generation in speed and quality, the unsatisfying performance of multi-view generation and the growing demands for higher mesh quality have led researchers to focus increasingly on the development of native 3D generation methods (Liu et al., 2023a; Gupta et al., 2023; Chen et al., 2024a; Wang et al., 2023b; Ren et al., 2024). 3DTopia trains

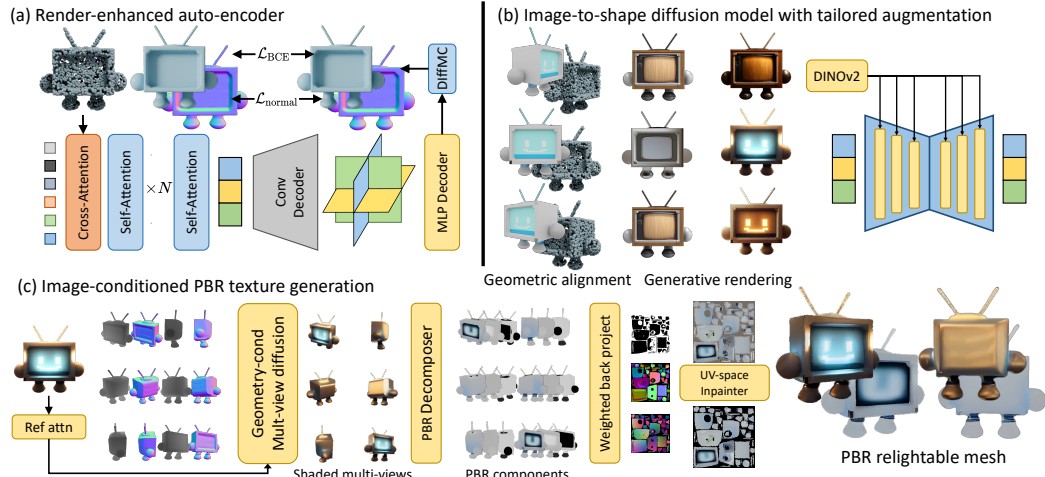

Figure 1: Overview of the proposed MeshGen. We first train a render-enhanced auto-encoder to compress 3D meshes to more compact latent space (Section. 3.1). We establish an image-to-shape diffusion model based on our tailored generative augmentations for improving image-shape alignment and generalization ability (Section. 3.2). The obtained mesh undergoes a reference attention-based multi-view synthesis and a PBR decomposer to obtain multi-view PBR channels. A UV-space inpainter is then exploited to fill the areas invisible in multi-view images (Section. 3.3).

a text-to-triplane NeRF diffusion model on pre-computed latents to achieve native text-to-3D generation. 3DShape2Vecset (Zhang et al., 2023a) and CLAY (Zhang et al., 2024a) exploit latent vector sets as representation, significantly enhancing the expressiveness of the latent space. CraftsMan (Li et al., 2024b) improves 3DShape2Vecset by incorporating point normal as input to the auto-encoder and proposes a normal enhancement process to generate finer details. Direct3D (Wu et al., 2024b) exploits triplane as latent representation for capturing the structural 3D information. Our method introduces a render-enhanced auto-encoder with geometric alignment and generative rendering augmentation during training, improving the performance of 3D native diffusion in image-to-3D tasks.

## 2.2 TEXTURE GENERATION

Initial efforts in mesh texturing have focused on only utilizing image diffusion models through iterative inpainting and optimization (Chen et al., 2023a; Jiang et al., 2024). TEXTure (Richardson et al., 2023) presents an iterative texturing method that employs a pre-trained depth-to-image diffusion model to progressively refine a 3D model's texture map from various views. TexFusion (Cao et al., 2023) enhances coherence by integrating texture information from multiple perspectives during the denoising stage. SyncMVD (Liu et al., 2023d) improves multi-view consistency by denoising in UV space and exploiting a self-attention reuse technique. FlashTex (Deng et al., 2024) proposes a light ControlNet for text-to-PBR generation. In addition to methods that use only image diffusion, various learning-based strategies initiate the training of generative texturing models using 3D textured mesh data (Nichol et al., 2022; Luo et al., 2023; Jun & Nichol, 2023; Li et al., 2022; Collins et al., 2022; Deitke et al., 2023; Chen et al., 2022; Yu et al., 2021b; Cheng et al., 2023). Texturify (Siddiqui et al., 2022) proposes a GAN-based pipeline with face convolution to colorize meshes without direct supervision. Point-UV (Yu et al., 2023) proposes a point diffusion to offer low-resolution global information and a UV diffusion for enhancing finer details. Paint3D (Zeng et al., 2023) proposes a coarse-to-fine strategy that firstly colorizes sparse views with depth-based inpainting and then improves texture quality within the UV space. Meta 3D TextureGen (Bensadoun et al., 2024) exploits a geometry-conditioned multi-view generator for text-to-texture generation.

## 3 METHOD

The overall pipeline of MeshGen is demonstrated in Fig. 1. We first train a render-enhanced auto-encoder to compress the 3D meshes into compact triplanes. A diffusion model is then established based on the proposed geometric alignment and generative rendering augmentation to en-

hance image-shape alignment and generalization ability. The decoded 3D mesh then undergoes our multi-view diffusion-based texturing pipeline for PBR material generation. The detailed MeshGen methodology is presented as follows.

## 3.1 RENDER-ENHANCED AUTO-ENCODER

**Transformer-based point-to-shape auto-encoder.** To compress the discrete 3D meshes into a continuous latent space, we adopted the same encoder as used in prior native 3D generation approaches (Zhang et al., 2023a; 2024a; Li et al., 2024b), namely the point-to-shape encoder. For a given 3D object, we first uniformly sample $N_P$ points from its surface. Following previous methods, we encode the sampled point cloud using Fourier positional encoding (Rahaman et al., 2019). Subsequently, a set of learnable queries is introduced to extract information from the point cloud through cross-attention, followed by a series of self-attentions to enhance the obtained representation. The complete encoding process can be formulated as

$$\mathbf{z} = \texttt{SelfAttn}^n(\texttt{CrossAttn}(Q, \texttt{FourierPE}(P))), \tag{1}$$

where $n$ refers to the number of self-attention layers, $\texttt{SelfAttn}$, $\texttt{CrossAttn}$ and $\texttt{FourierPE}$ represents self-, cross-attention, and Fourier positional encoding. Here $Q \in \mathbb{R}^{N_z \times d_z}$ and $P \in \mathbb{R}^{N_P \times 3}$ represent the learnable query set and the sampled point cloud respectively, $N_z$ and $d_z$ refer to the number of learnable queries and the dimension of the latent space. To incorporate render-based perceptual loss during auto-encoder training, we choose triplane as the latent representation (Wu et al., 2024b) instead of the latent vector set used in 3DShape2Vecset (Zhang et al., 2023a). This choice is motivated by the fact that when querying the occupancy, the latent vector set requires cross-attention with all latents, whereas the triplane only needs to pass through an MLP decoder, thus supporting surface extraction at a higher resolution. To obtain the occupancy of a specific point, a convolutional decoder is applied to upsample the encoded latent to a higher resolution to represent finer details. As analyzed in (Wang et al., 2023b; Wu et al., 2024b), we concatenate the three planes in the height dimension instead of the channel dimension to avoid artifacts caused by spatial misalignment. The occupancy of point $\mathbf{x}$ can be formulated as

$$\text{Occupancy}(\mathbf{x}) = \text{MLP}(\texttt{UpSample}(\mathbf{z}_{\text{tile}}), \mathbf{x}), \tag{2}$$

where $\texttt{Upsample}$ denotes the convolution-based upsampling network, MLP refers to the occupancy decoder, $\mathbf{z}_{\text{tile}}$ represents the height-concatenated triplane. As suggested in Odena et al. (2016), we use interpolation with convolution instead of deconvolution.

**Perceptual loss with ray-based regularization.** Previous point-to-shape auto-encoder relies solely on occupancy loss, the absence of perceptual loss leads to poor performance when reconstructing high-frequency details. In response, we propose supervising the auto-encoder using the rendered normal map. During training, we query the occupancy of a $256^3$ grid and extract iso-surface differentiably (Wei et al., 2023). To compute the render loss, we exploit nvdiffrast (Laine et al., 2020) to differentiably rasterize the normal map. However, we found in our early experiment that simply applying render loss alone will cause severe floaters in the final output mesh (see Fig. 8), which can also be observed in previous research (Wei et al., 2024). To address this issue, we propose a ray-based occupancy regularization that forces the occupancy in empty spaces to approach zero. As shown in the left part of Fig. 2, for each camera ray, we uniformly sample $N_s$ points between the ray-bounding box intersection and the surface point, enforcing their occupancy to be near zero by minimizing the sum of their occupancy. To save GPU VRAM and accelerate training, we interpolate the occupancy of the samples from the values used for previous surface extraction, rather than querying the triplane.

**Coarse-to-fine optimization.** Due to the locality of differentiable marching cubes, the gradients of the render loss can only propagate to points near surface vertices. Therefore, a coarse-to-fine training process is required to ensure the effectiveness of the render loss. Specifically, during the coarse stage, we apply the standard binary cross-entropy (BCE) loss for the point-to-shape auto-encoder, along with a KL loss to regularize the latent space and a total variation loss (Yu et al., 2021a) for reducing the floaters, i.e.

$$\mathcal{L}_{\text{coarse}} = \mathcal{L}_{\text{BCE}} + \lambda_{\text{KL}}\mathcal{L}_{\text{KL}} + \lambda_{\text{TV}}\mathcal{L}_{\text{TV}}, \tag{3}$$

where $\mathcal{L}_{\text{BCE}}$, $\mathcal{L}_{\text{KL}}$ and $\mathcal{L}_{\text{TV}}$ denote the BCE loss, the KL loss and the total variation loss respectively, $\lambda_{\text{KL}}$ and $\lambda_{\text{TV}}$ refers to the loss weights. After the coarse stage training, the model is capable of

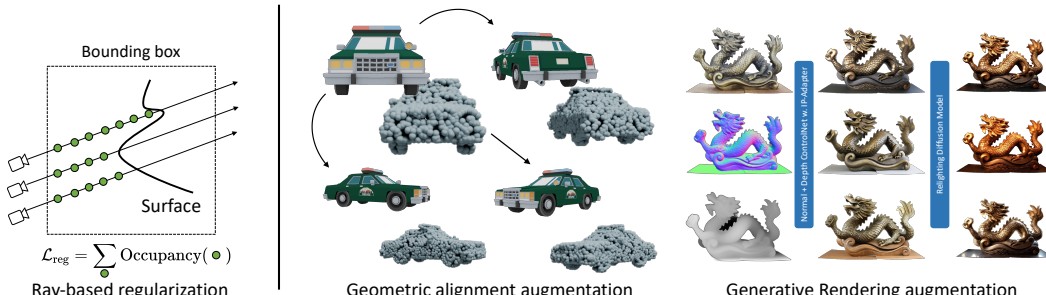

Figure 2: Illustration of the proposed ray-based regularization and two data augmentations.

reconstructing a coarse mesh from the input point cloud. In the refinement stage, we exploit render loss with ray-based regularization to enhance the details of the reconstructed mesh:

$$\mathcal{L}_{\text{refine}} = \mathcal{L}_{\text{BCE}} + \lambda_{\text{KL}}\mathcal{L}_{\text{KL}} + \lambda_{\text{TV}}\mathcal{L}_{\text{TV}} + \lambda_{\text{MSE}}\mathcal{L}_{\text{normal}}^{\text{MSE}} + \lambda_{\text{LPIPS}}\mathcal{L}_{\text{normal}}^{\text{LPIPS}} + \lambda_{\text{reg}}\mathcal{L}_{\text{reg}}, \qquad (4)$$

where $\mathcal{L}_{\text{normal}}^{\text{MSE}}$ and $\mathcal{L}_{\text{normal}}^{\text{LPIPS}}$ denotes the MSE and LPIPS (Zhang et al., 2018) loss for rendered normal, $\lambda_{\text{MSE}}$, $\lambda_{\text{LPIPS}}$, and $\lambda_{\text{reg}}$ refers to the corresponding loss weight. The concrete hyper-parameter settings are presented in appendix A.

### 3.2 Image-to-shape diffusion model with generative data augmentation

As shown in Fig. 4 and Fig. 5, compared to large reconstruction models, existing native 3D generation models almost always tend to generate symmetrical shapes that lack detail, leading to image-shape misalignment. We believe this phenomenon arises because the training of existing models predominantly relies on the Objaverse dataset (Deitke et al., 2022), and a considerable portion of the objects in Objaverse have symmetrical geometry and lack realistic textures. Therefore, the diffusion models trained on it tend to replicate simple geometries and struggle to generalize to images with complex textures or lighting. To optimize a diffusion model with strong generalization capabilities on limited data, we identify two key differences between the proposed pipeline and the previous NeRF-based native 3D generation pipeline (typical methods include Rodin (Wang et al., 2023b), 3DTopia (Hong et al., 2024), etc): **(1) Geometrical covariant auto-encoder.** Previous NeRF-based native 3D generation methods utilize a per-object optimized neural radiance field as the latent representation. This approach requires pre-computing and storing numerous latent vectors and lacks geometric covariance, as it necessitates re-optimizing the radiance field to obtain triplane latent variables after applying a transformation to the object. In contrast, our point-to-shape auto-encoder takes point clouds as input and does not require per-object optimization, it naturally achieves geometric covariance for transformations such as rotations by directly manipulating the point cloud. **(2) Appearance invariant image-to-shape modeling.** Previous methods learn to generate textured meshes from images, resulting in the entanglement of input images and the textures of the output meshes. In contrast, our diffusion model is specifically designed to map images to shapes, ensuring that the same shapes produce consistent renderings, regardless of variations in textures or lighting conditions. Based on both insights, we propose two data augmentations that are critical for training the image-to-shape model.

**Geometric alignment augmentation.** To enhance image-shape correspondence during training, we propose utilizing the geometric covariance property of our point-to-shape auto-encoder to ensure that different views of the same object correspond to different latents. Specifically, for each object in the dataset, we select one view from multi-view images as the condition and rotate the point cloud's azimuth to align the object's orientation with the selected image as the target (see the middle part of Fig. 2 for a simple demonstration). The aligned image-shape pairs are then used as training data for the diffusion model. Our experiments reveal that geometric alignment not only expands the training dataset but also significantly improves the alignment between generated shapes and images. We present the corresponding ablation study in Fig. 7.

**Generative rendering augmentation.** To enhance the generalization ability of the image-to-shape diffusion, we propose leveraging the appearance-invariant property by utilizing generative rendering to synthesize images with realistic textures and rich lighting based on the geometry of the object. Concretely, for each rendered image in the dataset, we utilize the corresponding normal map and

depth map as control signals to synthesize realistic renderings with ControlNet (Zhang et al., 2023b). To ensure that the augmented images do not deviate significantly from the originals, we inject the original image using an IP-adapter (Ye et al., 2023). We then use IC-light (Zhang et al., 2024b) to generate renderings under various lighting conditions and directions. Experiments show that generative rendering augmentation is highly beneficial for helping diffusion models understand lighting effects and generalize to real-world images (see Fig. 7 for the corresponding ablation study).

**Image-to-3D diffusion UNet.** We adopt a UNet similar to Stable Diffusion (Rombach et al., 2022) as the image-to-shape diffusion network. Following Rodin (Wang et al., 2023b), we concatenate triplanes along the height dimension as input to the UNet to avoid spatial mismatches. Interactions between different planes are handled via self-attention layers. To incorporate image information during diffusion, we encode the input image using DINOv2 (Oquab et al., 2024) and inject the extracted features into the denoising process through cross-attention. Following SD3 (Esser et al., 2024), we adopt rectified flow Liu et al. (2022) with lognorm timestep sampling as the training schedule. For more details on the training and inference of our diffusion UNet, please refer to appendix A.

### 3.3 TEXTURE GENERATION

#### 3.3.1 GEOMETRY-CONDITIONED MULTI-VIEW GENERATION WITH REFERENCE ATTENTION

Previous image-guided texturing pipelines (Richardson et al., 2023; Zeng et al., 2023; Perla et al., 2024) adopt IP-adapter (Ye et al., 2023) or personalization techniques (Ruiz et al., 2022; Gal et al., 2022) to inject the image to pre-trained diffusion models. These methods are unable to generate textures consistent with the original image and are highly prone to the multi-face problem.

To generate a Janus-free, image-consistent texture, we propose a geometry-conditioned ControlNet with reference attention to produce multi-view shaded images that align with the input in both appearance and lighting. Unlike the IP-adaptor, which maps images to prompts, reference attention (Zhang, 2023) integrates the keys and values from self-attention layers corresponding to the reference image into the denoising process, thus enhancing the consistency between the generated and original images. Our texturing model is based on Zero123++ (Shi et al., 2023a), which inherently uses scaled reference attention for generating multi-view images. To add



Figure 3: The effectiveness of the proposed reference attention fine-tuning.

geometry control, we trained a ControlNet (Zhang et al., 2023b) on top of the base model, enabling it to generate corresponding multi-view images from multi-view normal and depth maps. Specifically, our model mirrors the original ControlNet architecture but takes a six-channel image (3 for normal and 3 for depth) as input, i.e.

$$\mathbf{I}_i^{MV} = f_\theta(\mathbf{I}_{i-1}^{MV}, \mathbf{I}^{\text{front}}, i, h_\phi(\mathbf{I}^{\text{front}}, i | N^{MV}, D^{MV})), \tag{5}$$

where $\mathbf{I}_i^{MV}$ represents the multi-view images at denoising step $i$, $\mathbf{I}^{\text{front}}$, $N^{MV}$ and $D^{MV}$ refer to the input image, multi-view normal map and normalized depth map. However, applying the geometry-conditioned ControlNet directly to generated shapes yields unsatisfying results, especially when the input image and the multi-view normal and depth maps are not geometrically consistent, as shown in the middle part of Fig. 3. We attribute this degradation to the gap between training and inference. To mitigate this issue, we propose fine-tuning the reference attention layers to ensure that the generated results focus more on the semantic information of the reference image, rather than being overly sensitive to minor discrepancies. Specifically, we randomly apply slight translations to the condition images and perform rotations and scaling on the mesh to perturb the rendered depth and normals, thereby simulating situations of imperfect geometric consistency. We freeze the entire model except for the projection matrices of the reference attention layers and fine-tune it on the augmented dataset. As shown in Fig. 3, this lightweight fine-tuning effectively compensates for performance loss due to imperfect matching without compromising the model's generative capability. In contrast, full fine-tuned models produce overly smooth textures, and the original model suffers significantly from inconsistency.

Figure 4: Qualitative comparison with state-of-the-art large reconstruction models, including InstantMesh (Xu et al., 2024b), MeshLRM (Wei et al., 2024) and MeshFormer (Liu et al., 2024).

### 3.3.2 MULTI-VIEW PBR DECOMPOSITION

Previous methods that rely on pre-trained image diffusion models generate images with inherent shading effects, leading to lighting-baked-in textures. In order to generate relightable PBR textures, we propose a diffusion-based multi-view PBR decomposer, aiming to decompose the shaded multi-view image to corresponding intrinsic channels with multi-view information. Specifically, inspired by Zeng et al. (2024), our PBR decomposer employs an InstructPix2Pix (Brooks et al., 2023)-based architecture, concatenating the shaded image latent and the noisy latent along the channel dimension to output the desired PBR components, i.e.

$$\mathbf{I}_i^{MV}(y) = g_\phi(\mathbf{I}_{i-1}^{MV}(y); \mathbf{I}^{MV}, \mathbf{I}^{front}, i, \tau(y)), \tag{6}$$

where $g_\phi$ represents the denoising UNet, $\tau$ denotes the CLIP (Radford et al., 2021) text encoder, $y \in \{\texttt{"metallic"}, \texttt{"roughness"}, \texttt{"albedo"}\}$ denotes the component prompt for PBR texture, $\mathbf{I}_i^{MV}(y)$ refers to the denoised $y$ component at timestep $i$. After generating multi-view PBR components, we use a view-weighted approach to fuse the multi-view textures in UV space, i.e. $\text{UV} = \sum_i \text{Softmax}_i(\text{BP}(\mathbf{I}^{(i)}), \text{BP}(w^{(i)}))$ , where BP refers to back-projecting the rendered image to UV space, $\mathbf{I}^{(i)}$ denote the target image for the $i$-th view and $w^{(i)}$ represents the pixel-wise weight calculated as the cosine of the viewing angle to the point.

### 3.3.3 UV-SPACE TEXTURE INPAINTING

For meshes with complicated topology, the generated views are not adequate to cover the entire surface of the mesh. We propose a UV-space texture inpainter to fill the invisible part of the multi-views. Specifically, due to the significant gap between casual images and texture maps, we first train a LoRA on the texture maps in Objaverse with "A UV space [y] texture map of [*]" as textual prompt, where the [*] represents the original caption of the corresponding 3D object generated using Cap3D (Luo et al., 2023) and [y] represent the PBR component prompt. Subsequently, we merge LoRA into the original UNet and train an inpainting ControlNet on top of it. To let the

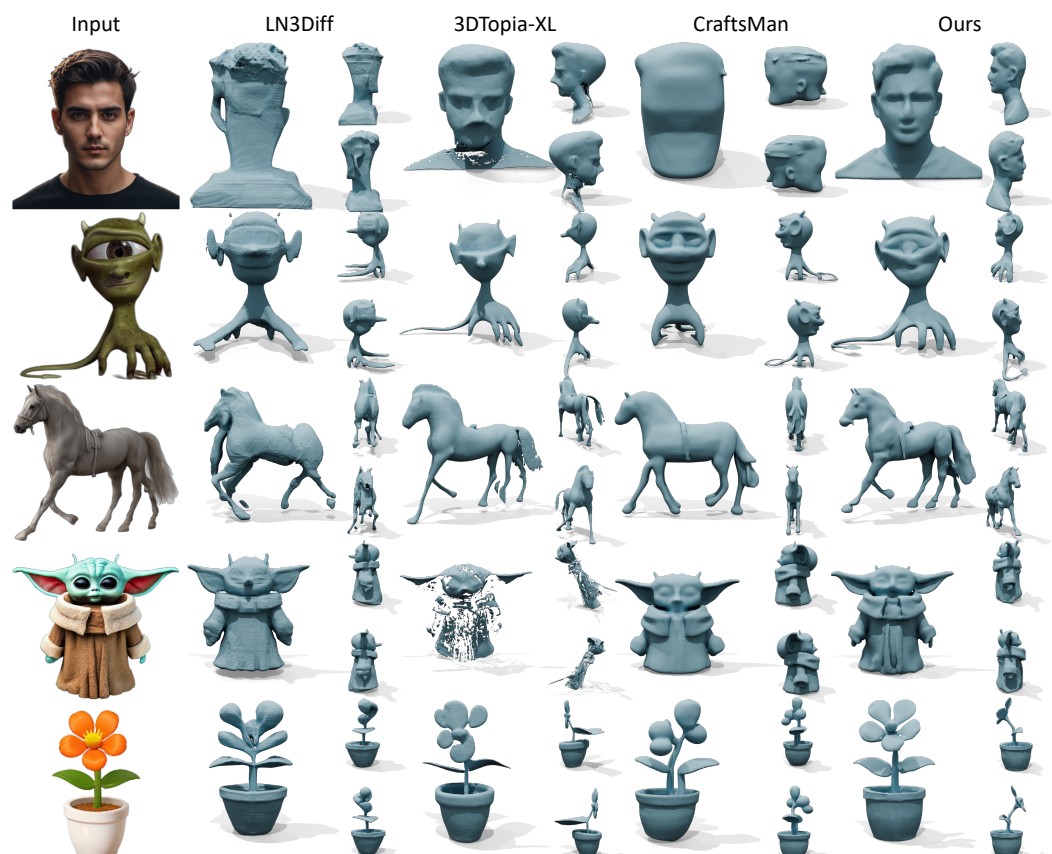

Figure 5: Qualitative comparison with state-of-the-art native 3D diffusion models, including Crafts-Man (Li et al., 2024b), LN3Diff (Lan et al., 2024), and 3DTopia-XL (Chen et al., 2024a).

2D image perceive information from the original mesh, our control signal includes not only the masked image but also the normal and position maps in UV space. For the mask setup during training, we simulate the inference process by back-projecting the visible mask from the fixed views of our multi-view diffusion into UV space to obtain the invisible mask. To enhance robustness, we randomly erode the visible masks from multi-views. We present more details about the texturing pipeline in appendix A.3.

# 4 EXPERIMENTS

## 4.1 MESH GENERATION

In our experiments, we compare our method with state-of-the-art image-to-3D methods from the following two categories.

**Large reconstruction models.** (Hong et al., 2023; Li et al., 2023a;b; Wang et al., 2024; Tang et al., 2024a) exploit a neural network to map sparse views into 3D representations. We compare the proposed methods with recent state-of-the-art large reconstruction models, including InstantMesh (Xu et al., 2024b), MeshLRM (Wei et al., 2024) and MeshFormer (Liu et al., 2024). To be clear, the results of MeshLRM and MeshFormer are obtained through their official demo, since their source code is not publicly available.

**Native 3D generation models.** (Liu et al., 2023a; Zhang et al., 2023a; 2024a) compress discrete 3D meshes into a continuous and compact latent space using a 3D auto-encoder, followed by a diffusion model trained on this latent space to achieve 3D generation. We compared our method with recent

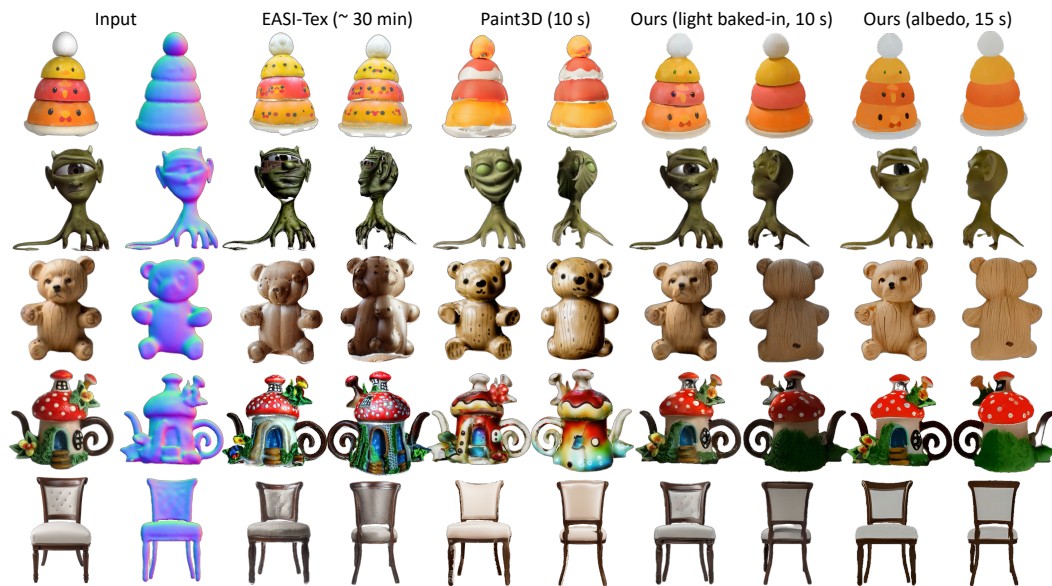

Figure 6: Qualitative comparison with image-guided mesh texturing pipelines, including EASI-Tex (Perla et al., 2024) and Paint3D (Zeng et al., 2023).

state-of-the-art open-source models, including LN3Diff (Lan et al., 2024), 3DTopia-XL (Chen et al., 2024a), and CraftsMan (Li et al., 2024b).

We present a qualitative comparison between our method and large reconstruction models in Fig. 4. Our approach significantly outperforms others in generating high-quality geometry. Specifically, our method excels at producing complex structures from images, such as backpack handles, the mouth of the alien, and the file sorter, where large reconstruction models struggle. Additionally, large reconstruction models often suffer from poor multi-view generation outcomes, resulting in lower quality when reconstructing details that require multi-view consistency, such as the gap between an eagle's legs. In Fig. 5, we provide a qualitative comparison with other 3D native generation methods, where our approach significantly outperforms the others. We observe that previous methods often produce symmetrical and overly simplistic geometric structures, resulting in noticeable misalignment with the input images. Our method, leveraging the proposed two data augmentations, effectively addresses this issue. This image-shape alignment enhances the controllability of shape generation and simplifies subsequent texturing. For 3DTopia-XL, its explicit representation results in a lower compression rate compared to the point-to-shape auto-encoder. Consequently, despite higher training costs, MeshGen still outperforms 3DTopia-XL by a large margin.

## 4.2 TEXTURE GENERATION

As there exist no algorithms specifically designed for image-consistent texturing, we compared our method with state-of-the-art image-guided approaches, including EASI-Tex (Perla et al., 2024) and Paint3D (Zeng et al., 2023). Fig. 6 demonstrates the texturing results on meshes generated by our image-to-shape diffusion model. Our method significantly outperforms previous approaches in quality and texture-image consistency, even when the shape and image do not perfectly align. In contrast, despite using image inversion which requires additional per-input optimization, EASI-TEX still struggles to maintain consistency with the original image and takes dozens of times longer than our approach. Paint3D, which uses simple back projection and inpainting, exhibits noticeable seams in the generated textures and is prone to the Janus problem. We further showcase the PBR materials generated by our method in Fig. 13, highlighting its remarkable capability in handling objects with complicated appearances under different lighting conditions.

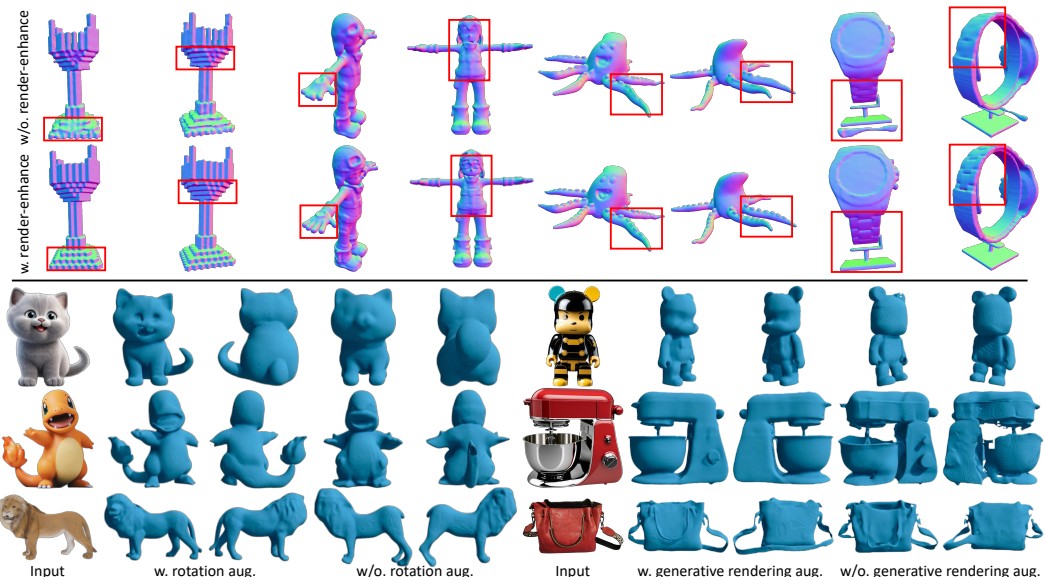

Figure 7: Ablation study results on render-enhanced auto-encoder, geometric alignment augmentation, and generative rendering augmentation.

## 4.3 ABLATIONS

**Render-enhanced auto-encoder.** To evaluate the significance of incorporating render loss in the point-to-shape auto-encoder, we trained a variant without this component and compared it to our render-enhanced version, as illustrated in the upper part of Fig. 7. The render-enhanced auto-encoder exhibits markedly superior performance, particularly in capturing high-frequency details such as the suckers on tentacles and the gaps in watch bands.

**Geomtric alignment augmentation.** To validate the impact of geometric alignment augmentation, we trained a smaller diffusion UNet consisting of 4 layers without this augmentation for 300 epochs as an ablation study. The comparison is presented in the lower left part of Fig. 7. Evidently, the diffusion model trained without geometric alignment augmentation tends to generate symmetric objects, whereas our model produces shapes that align well with the input images, significantly enhancing the model's controllability.

**Generative rendering augmentation.** To assess the impact of generative rendering augmentation on image-to-shape diffusion training, we trained a smaller model without this augmentation, as depicted in the lower right part of Fig. 7. The model trained without generative rendering augmentation exhibits poor performance in handling lighting effects in images and struggles to infer the geometric structure based on lighting cues, such as determining the shape of the doll's head and the shape of the blender. These findings suggest that generative rendering augmentation significantly enhances the model's ability to understand lighting effects and interpret real-world images.

More ablations regarding to auto-encoder, image-to-shape diffusion model, and texture generation model are presented in appendix B.1.

## 5 CONCLUSION

In this paper, we propose MeshGen, a novel pipeline for generating delicate PBR textured mesh given a single image. MeshGen encodes 3D meshes to compact latent space with a render-enhanced auto-encoder. Based on our in-depth analysis of point-to-shape auto-encoder and image-to-shape diffusion, we propose to train the diffusion model with geometric alignment and generative rendering augmentation to address the issues of image-shape misalignment and poor generalization ability. Besides, to generate PBR texture consistent with the image, we establish a reference attention-based multi-view generator followed by a PBR decomposer to obtain PBR components and a UV-space inpainter to fill the invisible part. Extensive experiments have demonstrated the effectiveness of our method. We hope our work will aid in a deeper understanding of native 3D diffusion and provide support for future related research.

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

# A IMPLEMENTATION DETAILS

## A.1 AUTO-ENCODER

We present our hyper-parameter setting in training auto-encoder in Tab. A.1.

Table 1: Concrete hyper-parameter setting of our render-enhanced auto-encoder.

| Symbol | Meaning | Value |
|---|---|---|
| $N_P$ | Number of points sampled from a mesh | 65536 |
| $N_z$ | Number of learnable queries | 3072 |
| $n$ | Number of self-attention layers | 10 |
| $d_z$ | Dimension of the latent space | 16 |
| $N_s$ | Number of samples for calculating ray-based regularization loss | 128 |
| $\lambda_{\text{KL}}$ | Loss weight for KL loss | $10^{-6}$ |
| $\lambda_{\text{TV}}$ | Loss weight for TV loss | $5 \times 10^{-3}$ |
| $\lambda_{\text{MSE}}$ | Loss weight for normal MSE loss | 1.0 |
| $\lambda_{\text{LPIPS}}$ | Loss weight for normal LPIPS loss | 2.0 |
| $\lambda_{\text{reg}}$ | Loss weight for ray-based regularization loss | 0.5 |

We first train our auto-encoder for 150 epochs in the coarse stage with a batch size of 192. The model obtained after the coarse stage can reconstruct the rough shape of the original mesh but lacks details. We then train the auto-encoder for another 50 epochs with the proposed render loss and a batch size of 16.

## A.2 IMAGE-TO-SHAPE DIFFUSION MODEL

### A.2.1 DATA AUGMENTATION

In generative rendering data augmentation, to enhance the similarity between the generated images and the original image, in addition to using the normal depth ControlNet and IP-adapter, we set the initial noise to the latent of the original image with maximum noise added. For relighting diffusion, we used IC-light (Zhang et al., 2024b). Specifically, during data augmentation, we randomly select one lighting direction from the pre-defined light initial latent in IC-light (i.e., uniformly select from left, right, top, and bottom), and choose one lighting condition from a set of predefined light prompts.

### A.2.2 IMAGE-TO-SHAPE DIFFUSION MODEL

Our diffusion UNet takes in the noised triplane latent and exploits 8 ResNet blocks with spatial self-attention as the encoder and a symmetric architecture as the decoder. We exploit DINOv2-G (Oquab et al., 2024) to encode the input image and inject the extracted feature to the diffusion UNet using cross-attention. For the diffusion schedule, we follow SD3 to use the simple yet effective rectified flow Liu et al. (2022) with timesteps sampled from a standard logit-normal distribution. We train the image-to-shape diffusion model with our proposed augmentations on a filtered subset of GObjaverse (Qiu et al., 2024), which consists of about 120k high-quality multiview-mesh pairs. To handle input images with different elevations, since the meshes in Objaverse are aligned in the gravity axis, we force the diffusion to generate meshes with absolute elevation equal to zero. We experimentally found that this conditioning method works better than generating meshes with a rotation in elevation, as suggested in Chen et al. (2024c). We train the diffusion UNet of 16 NVIDIA A800 GPUs using bf16 precision with an effective batch size equal to 1536. The whole training lasts for about 18 days.

## A.3 PBR TEXTURE GENERATION

**Data preparation.** To train the geometry-conditioned ControlNet and the multi-view PBR decomposer, we rendered multi-view images and corresponding multi-view normals, depth, albedo, roughness, and metallic maps of a subset of Objaverse containing PBR materials using Blender.

| w. regularization | w/o. regularization | w/o. depth filtering | w. depth filtering | w/o. UV inpainting | w. UV inpainting |

Figure 8: Ablations on ray-based regularization, depth filtering, and UV space inpainting.

This constitutes a dataset comprising 35k multi-view images. For UV space inpainting, we calculate the multi-view visible masks and back-project them into UV space to determine the invisible part of the texture map and render the UV space position and normal map. To further enhance robustness, we randomly erode the visible mask in pixel and UV space.

**Geometry-conditioned ControlNet training.** To ensure the model perceives precise depth and positional information, we did not transform depth to normalized disparity as done in the original depth ControlNet (Zhang et al., 2023b); instead, we performed a unified multi-view normalization based on camera distance and object bounding box. Specifically, the depth map is processed as $D_{\text{normalized}} = \frac{D - \text{bias}}{\text{scale}}$, where bias equals to camera distance minus the length of the diagonal of the bounding box (i.e. the minimal possible depth value) and the scale equals to the length of the diagonal of the bounding box.

**Multi-view target back-projection.** As detailed in the main text, the obtained multi-view PBR components are merged in UV space using back-projection with softmax. We apply a softmax operation with a temperature of 0.1 to ensure consistent textures. However, images generated by ControlNet sometimes extend beyond object boundaries, causing some pixels to be back-projected onto surfaces behind them, leading to artifacts. To address this, we propose a simple depth filtering technique. For each view, we identify locations in the depth map where sudden changes occur and exclude these pixels during back-projection. Our experiments demonstrate that this approach effectively reduces artifacts, and the color values of the corresponding surface points can be supplemented by other views, as shown in the middle of Fig. 8.

**UV space inpainting.** Our UV space inpainter is a multi-channel ControlNet trained on top of the LoRA fine-tuned diffusion model. The input to our inpainting model is a 9-channel image: the first three channels represent the normal map in UV space, the middle three channels represent the position map, and the last three channels contain the masked texture map, with pixel values set to -1 in regions that requires inpainting. During inference, we follow ControlNet inpainting (Zhang et al., 2023b), applying masking in the latent space to maintain consistency in areas that do not require inpainting.

## B MORE EXPERIMENTS

### B.1 MORE ABLATIONS

**The effectiveness of ray-based regularization.** We show in the left part of Fig. 8 an example obtained using an auto-encoder trained without ray-based regularization. Without ray-based regularization, the training of the auto-encoder quickly became unstable, resulting in severe floaters in the reconstructed mesh.

**Quatitative ablation study on render-enhanced auto-encoder.** To better assess the importance of incorporating render loss in our render-enhanced auto-encoder, we propose several variants and demonstrate the corresponding accuracy and volumeIoU on a validation set of Objaverse consisting of 2048 objects in Tab. B.1. Here, "base" represents the case with only BCE loss, while "w/. 3D GAN loss" represents incorporating the 3D patch-based GAN loss proposed in Zheng et al. (2022).

As shown in Tab. B.1, removing either the MSE loss or the LPIPS loss leads to a certain performance drop. Moreover, compared to the 3D patch-based GAN loss, the proposed render-based perceptual loss is more beneficial for auto-encoder training.

**UV-space texture inpainting.** In the right part of Fig. 8, we compare the mesh obtained without UV inpainting. The figure clearly shows that without UV inpainting, colors may be missing from re-

Table 2: Quantitative ablation study on the proposed render-enhanced auto-encoder.

| Setting | Accuracy↑ | VolumeIoU↑ |
|---|---|---|
| Ours | **96.987** | **91.045** |
| w/o. $\mathcal{L}_{normal}^{MSE}$ | 95.972 | 89.977 |
| w/o. $\mathcal{L}_{normal}^{LPIPS}$ | 96.021 | 90.044 |
| w/. 3D patch GAN loss | 96.224 | 90.149 |
| base | 94.745 | 87.164 |

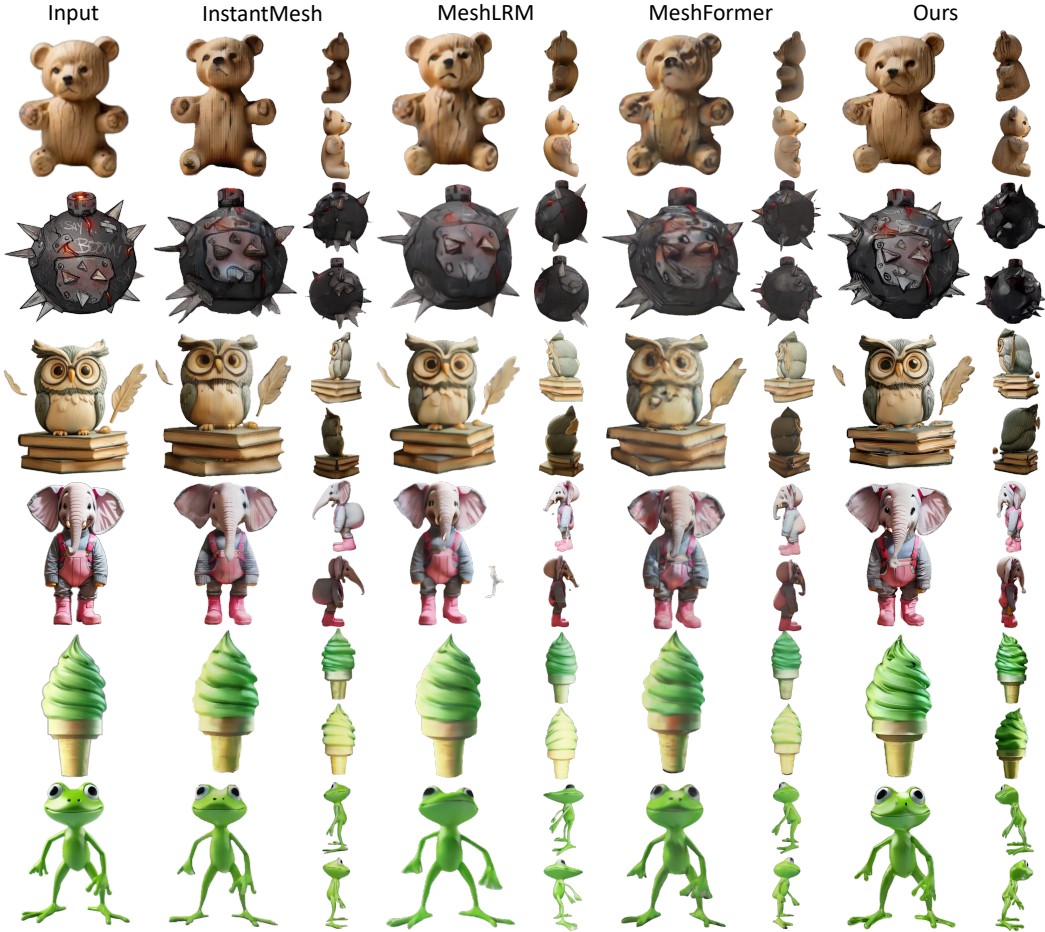

Figure 9: Qualitative comparison on textured meshes with state-of-the-art large reconstruction models, including InstantMesh (Xu et al., 2024b), MeshLRM (Wei et al., 2024) and MeshFormer (Liu et al., 2024).

gions not visible from the fixed viewpoints generated by multi-view diffusion. UV space inpainting effectively fills these regions with appropriate colors, enhancing both the visual quality and realism of the model.

## B.2 MORE RESULTS

**Compare with large reconstruction models with texture.** To comprehensively compare our approach with large reconstruction models, we compare the final generated textured mesh in Fig. 9. It is evident from the figure that our method not only exceeds the previous best large reconstruction models in geometry but also produces clearer and more consistent textures.

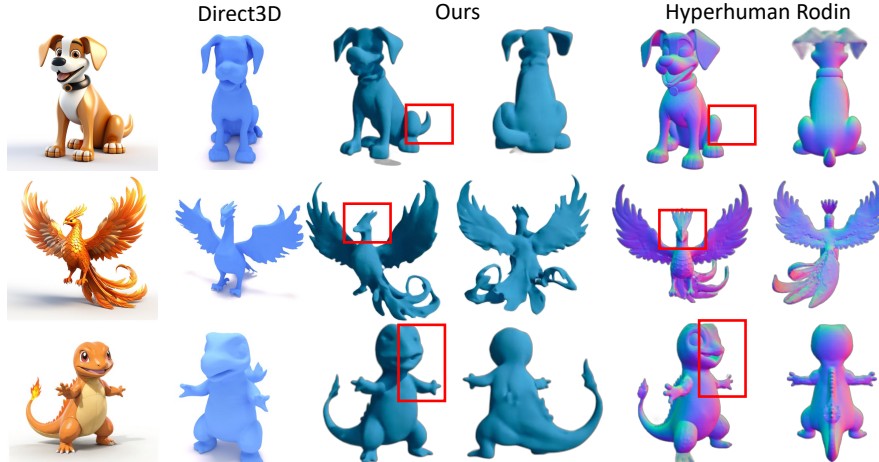

Figure 10: Comparison with non-open-source commercial products, including Direct3D and Hyperhuman Rodin.

**Compare with commercial products.** In Fig. 10, we compare our method with existing non-open-source commercial products. The results for Direct3D are sourced from their paper, while those for HyperHuman Rodin are generated on their official website without the "symmetric" tags. Although our method is currently limited by lacking high-quality data and computational resources, resulting in slightly lower mesh quality compared to commercial products, our proposed augmentation allows for better alignment with the images. We believe that with increased computational power and more high-quality data, our method can match the mesh quality of commercial products while preserving image-shape alignment.

**Real-world images.** To validate the performance of our method on real-world objects, we present a set of textured meshes generated from casual captures in Fig. 11. As shown in Fig. 11, our method is capable of generating reasonable shapes and consistent textures when processing real objects, demonstrating the generalization ability of our pipeline.

**PBR decompositions.** In Fig. 13, we present the intrinsic channels estimated using our proposed multi-view PBR decomposer. The results show that our PBR decomposer can accurately infer the PBR materials of objects by leveraging multi-view information and can still generate multi-view consistent results under complex lighting conditions.

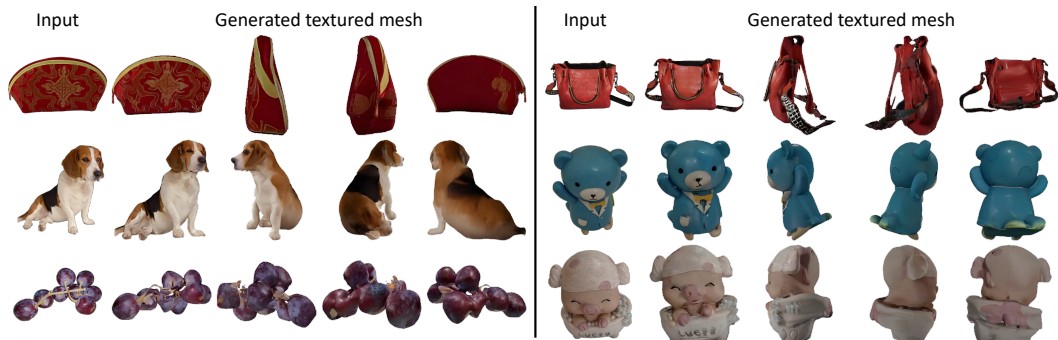

Figure 11: Performance of MeshGen on real-world captures.

## C    LIMITATIONS

Although our method has made some progress in native image-to-3D generation, there are still limitations in the following three areas.

1. Due to the limited resolution of multi-view diffusion generation and the constraints of the auto-encoder used, our texture model struggles to accurately reproduce high-frequency de-

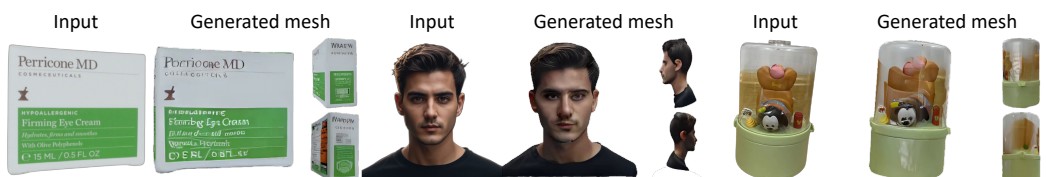

Figure 12: Some typical failure cases of MeshGen.

tails, such as the text on the box in the left part of Fig. 12. We believe that using more advanced network architectures could achieve higher-resolution multi-view generation.

2. Our texture model finds it challenging to accurately capture textures and lighting effects from input images when dealing with objects with complex high-frequency information and lighting conditions, as shown by the face in the center of Fig. 12.

3. Our geometry generation model currently cannot effectively handle transparent objects, as illustrated by the object on the right in Fig. 12.

Addressing these limitations will be the focus of our future research.

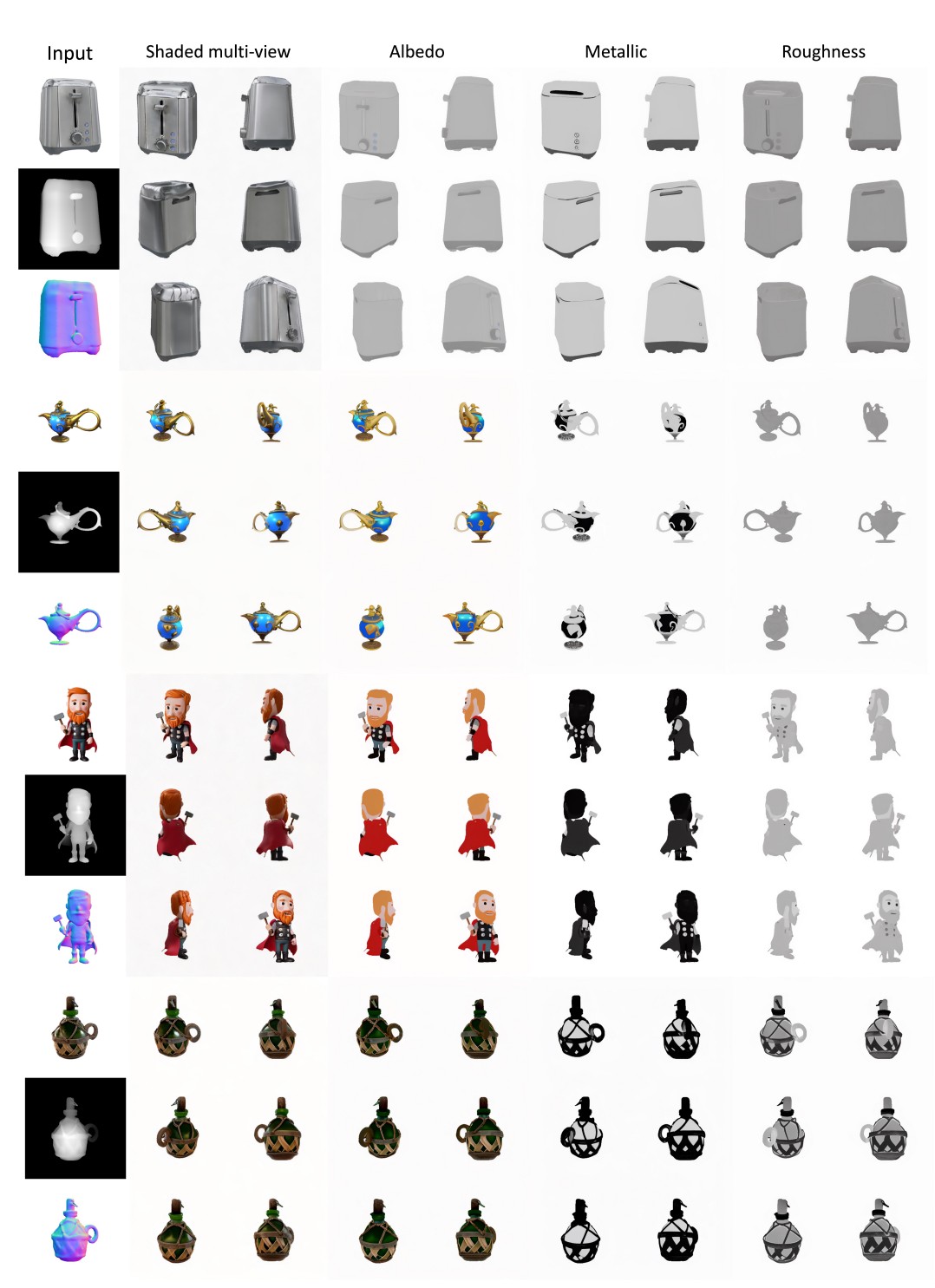

Figure 13: Intrinsic channels estimated using our multi-view PBR decomposer. The proposed PBR decomposer can handle images with complicated material under different lighting conditions.

