# OpenReview forum: "MeshGen: Generating PBR Textured Mesh with Render-Enhanced Auto-Encoder and Generative Data Augmentation"
_ICLR.cc/2025/Conference — ICLR 2025 Conference Withdrawn Submission_

### Official Review · Reviewer_rYth · 2024-11-01

**Soundness:** 3
**Presentation:** 3
**Contribution:** 3
**Rating:** 6
**Confidence:** 4

**Summary:**

The paper introduces a novel pipeline, termed MeshGen, designed for the generation of high-quality 3D meshes with physically based rendering (PBR) textures from a single image.
During the geometry generation stage, the authors first utilize a render-enhanced auto-encoder to encode 3D meshes into a compact latent space. Subsequently, an image-to-shape diffusion model is trained, incorporating geometric alignment and generative rendering augmentation to address challenges related to image-shape misalignment and the model's generalization capability.
In the texture generation stage, the paper establishes a reference attention-based multi-view generator, which is subsequently followed by a PBR decomposer to extract PBR components, along with a UV-space inpainter to complete the rendering of occluded areas.

**Strengths:**

1. Generative rendering augmentation appears to hold significant promise.

2. The results of geometry generation demonstrate satisfactory performance relative to available open-source non-commercial methods.

3. The outcomes of PBR material generation are both compelling and credible, particularly the examples of metal objects presented in the appendix.

**Weaknesses:**

1. Robustness of PBR material decomposition? During the texture generation stage, the method initially produces multi-view shaded images, which are subsequently decomposed into their PBR components. However, the variability in light and shadow effects within these shaded images can be substantial. I am particularly interested in the robustness of the PBR decomposer. Specifically, I am curious to know whether the decomposer can effectively manage scenarios involving more intricate lighting conditions.

2. PBR material generation results on more complex metal objects? The paper has demonstrated promising PBR material generation results on various metal objects, including a teapot and a roaster. I am intrigued by the potential outcomes of PBR generation on more complex metal objects, such as a game asset axe or a detailed representation of Iron Man.

**Questions:**

1. Geometry comparison with commercial products? I acknowledge that lower mesh quality, attributable to the constraints of high-quality data and computational resources, is a reasonable compromise. However, I am intrigued by your rationale for deeming the alignment termed "symmetry" as unnecessary.

2. PBR material generation from scratch V.S. PBR material decomposition from shaded images? The paper presents a promising approach involving multi-view RGB image generation and subsequent multi-view RGB-to-PBR decomposition. Concurrently, an alternative methodology exists for generating albedo, metallic, and roughness attributes from scratch, as exemplified by the HyperHuman Rodin (CLAY). I am curious about your decision to opt for PBR material decomposition over this alternative technique. Additionally, I am interested in your assessment of the strengths and weaknesses of these two techniques (discussion without qualitative results is acceptable since HyperHuman Rodin is not open-source).

3. More details about reference attention? Although I am not familiar with the reference attention, I am quite positively impressed by it. Could you please provide more detail?

---

### Official Review · Reviewer_UCjb · 2024-11-02

**Soundness:** 3
**Presentation:** 2
**Contribution:** 2
**Rating:** 5
**Confidence:** 4

**Summary:**

The authors propose PBR textured mesh reconstruction method from single view image.

The model has several components, which are render-enhanced auto-encoder, image-to-shape diffusion model with augmentations, and image-conditioned PBR texture generation.

Compared to the previous methods, the paper shows enhanced and fine-grained textured mesh generation results.

**Strengths:**

The paper provides a thorough literature survey of previous studies, analyzes the weaknesses of these methods, and presents a concrete training model and pipeline.

The qualitative results are improved than to the previous methods, especially for the fine-grained details and well-presenting to the given input view images. The human head results in Fig. 5 is the promising result, because it is out domain of the object dataset (especially for the Objaverse). It will be better to add more human head (out-domain) results.

The paper can deal with PBR textures which is essential to practical applications.

**Weaknesses:**

The figures of the pipeline and modules are hard to understand and make less intuitive because of heavy abbreviation, especially for Fig. 1, Fig. 2, and Fig. 4.

It is hard to understand the whole pipeline process of the paper’s modules. Especially for the Fig. 1, the reviewer wonders what is the connection between (a) render-enhanced auto-encoder and (b) image-to-shape diffusion model with tailored augmentation. It is nice to have a well thought out and fleshed out design, but is is hard to understand the connectivity of the entire module.

For the Fig. 4 and Fig.7, it is better to denote which is the paper’s method (ours).

The paper lacks the discussion of limitations.

**Questions:**

What is the detailed training recipes about training dataset, GPU specs, and training time.

The paper needs to show qualitative results with unseen evaluation datasets, such as Google Scanned Objects (GSO) datasets to a stronger rationale for the improved results. Only the qualitative result of the ablation study is given.

For the geometric alignment augmentation, the reviewer wonders why it is the augmentation. In L. 261-263, the authors say that “we select one view from multi-view images as the condition and rotate the point cloud’s azimuth to align the object’s orientation with the selected image as the target”. This seems to be just training with an image corresponding to each multi-view angle for a given point cloud, but it's hard to understand why this is an augmentation. The reviewer wonders if training to correspond to multi-view images for a given 3D point cloud is not an existing method for multi-view learning, and what is added differently by augmentation.

For the multi-view PBR decomposition, are the generated images between different PBR components of the same view image, and multi-view images consistent?

---

### Official Review · Reviewer_6QQ6 · 2024-11-04

**Soundness:** 2
**Presentation:** 3
**Contribution:** 3
**Rating:** 6
**Confidence:** 5

**Summary:**

The authors introduce solution to image-to-3D generation problem. Instead of predicting textured mesh, they design native 3D generation approach, with a separate PBR texturing stage.

* 3D generation stage consists of training a render-enhanced point-to-shape auto-encoder; they chose triplane representation instead of previously used vector set for rendering efficiency reasons.
* Following the autoencoder training stage, they employ a diffusion UNet on top of h-stacked triplane features, with cross-attention on input image DINOv2 features
* For texture prediction, authors train a multiview ControlNet, applied on top of Zero123++ to predict the multiview shaded renders, with another Instruct-pix2pix decomposer to separate it into PBR materials.

**Strengths:**

* Authors provide a comprehensive image-to-3D pipeline (notably, based on 3D diffusion), which performs on par or better than LRMs, and far better than other native 3D methods
* The paper is well-structured, clear, and easy to follow
* A key strength is that the authors correctly point out a common issue with 3D latent diffusion models, where the outputs often look symmetric. They visually prove that geometric alignment augmentation is a well-suited solution for this problem. This is original and significant contribution of the paper.
* Generative rendering augmentation is shown to be an effective augmentation pipeline in practice. This idea, to my knowledge, is original.

**Weaknesses:**

* The paper lacks any quantitative evaluation, particularly in two key areas:
  * autoencoder quality: This could be benchmarked against models like 3DShape2VecSet.
  * (biggest weakness) 3D reconstruction quality (geometry-only): Without quantifiable metrics, it's difficult to assess the quality claims. Comparisons with recent LRM papers such as TripoSR and Stable Fast3D on datasets like GSO or OmbiObject3D would be beneficial.

* The proposed render-enhanced autoencoder feels more like a combination of existing methods (e.g., 3DShape2VecSet + render-based loss, a technique used in prior works like DMV3D) rather than a novel contribution.

* I find the justification for ray-based regularization questionable. The paper mentions that
> render loss alone leads to severe floaters

  Isn't that what the BCE loss on occupancy is meant to address? If used correctly, BCE should perform as well as, if not better than, ray-based regularization.

* The generative rendering augmentation seems like a training trick to artificially boost dataset diversity. While this may improve performance, it could complicate future comparisons. I'd recommend reporting metrics without this augmentation for a clearer evaluation.

* Finally, the texturing pipeline appears to be a technical application of existing ideas and seems more complementary to the paper’s core contribution.

In summary, while the paper lacks core significant novelty, it presents a well-executed combination of techniques for image-to-3D problem with native 3D diffusion models.

**Questions:**

* Please include quantitative comparisons with the latest LRM papers (e.g., TripoSR, Stable Fast3D) on datasets like GSO or OmbiObject3D.

I understand that you're training a native 3D diffusion model, which is inherently more complex and resource-intensive compared to LRMs. It would be helpful to discuss whether your results are constrained by resource limitations, and to what extent techniques like rendering augmentation were necessary to achieve your results.

---

### Official Review · Reviewer_JCJb · 2024-11-05

**Soundness:** 2
**Presentation:** 2
**Contribution:** 1
**Rating:** 5
**Confidence:** 4

**Summary:**

Building large models for translating images to 3D models is now a very hot topic. This work is a follow-up in this direction. As mentioned, three things are new: 1) incorparating render-based perceptual loss into the auto-encoder training; 2) two augmentation stratigies are proposed; 3) a texturing pipeline with reference-based attention mechanism is presented. The experiments validate the effectiveness of the proposed designs.

**Strengths:**

- The paper is easy to read.
- The results in all figs look good, when comparing with existing methods, especially for the geometric details.

**Weaknesses:**

My concerns include:
- It lacks training details, especially the data. What is the scale of the training 3D models? Is it same with meshLRM, meshFormer and Craftsman? If no, the comparison with them may be not fair.

- It lacks quantitative anlaysis of the image-to-shape models. Currently, there are only some selected examples are shown which is not enough to support the claim of SOTA accuracy.

- The paper is not well-motivated. What kind of issues does this paper aim to address? This is not clear to me.

- Lack technical insights. Involving of render-based perceptual loss, the propose new augmentation strategies, attention-based texturing pipeline etc. All the claimed new things are some engineering methods. I believe these can improve the performance, however it cannot brings the community new insights.

**Questions:**

No

---

### Official Review · Reviewer_EXPc · 2024-11-06

**Soundness:** 1
**Presentation:** 2
**Contribution:** 2
**Rating:** 3
**Confidence:** 3

**Summary:**

The paper studies the problem of recovering 3d and some material properties of the objects studied from an image (or from multiple images, or from multiple images and their normal maps.  Which of these is correct was not clear from the paper).
Extensive experimentation was performed to optimize the terms of various loss functions in order to give high-quality visual results of the re-rendered captured shapes.  Extra care was taken to ensure that physically-based rendering of the captured surfaces allowed for the captured objects to be rendered under different lighting conditions.

The paper operated within a subset of the Objaverse dataset, consisting of 35k multi-view images.

**Strengths:**

The final images do indeed look better than the rendered comparison images, for the images shown.

**Weaknesses:**

There aren't numerical results given for the system performance.  Thus, the reader is constantly questioning, "are these results cherry picked"?
The paper is written for an audience of researchers operating in the same sub-field:  people reconstructing 3d, training and testing of Objaverse dataset images.  I'm not in that set of researchers (and ICLR readers generally won't be) and many aspects of the paper were unclear to me, see the questions below.
This is an engineering paper, a paper showing how to tweak parameters
to achieve slightly better results in a very crowded field.  As I read
the paper, I kept asking myself, "what do I learn from this?"  and I
rarely came up with an answer to that question.  The message is,
extensive parameter tweaking results in slightly better performance.
I don't feel that's a message that we need to convey to the ICLR
audience.

My concerns with the paper:
(1) There's no high-level story presented, no obvious set of take-aways that the reader learns.
(2) The paper doesn't present the work in a way that's accessible to readers outside of this particular subfield.
(3) Quantitative performance evaluations are not given, just lots of thumbnail images.  This is unsatisfying, and not persuasive, since the reader wonders about bad results not being shown.  If the results are indeed a random selection of the system outputs, please say so.
(4) Generalization beyond the one dataset trained on was relegated to one figure in the appendix.  Same with failure cases.

**Questions:**

please tell me the line number where you state:  what the input is to the system, ie, how many views are assumed to be input?  Are surface normals also assumed to be input?
If you both train and test on the Objaverse dataset, then why can't you report more quantitative measures of performance?  Presenting small thumbnails as the research output leaves the reader wondering if the results we're viewing are just the examples that happened to work well.

---

### Note · Authors · 2024-11-15

I have read and agree with the venue's withdrawal policy on behalf of myself and my co-authors.